# Dosimetry study of 3D-printed noncoplanar template-assisted CT-guided [125]I seed implantation for the treatment of recurrent and metastatic tumors in the head and neck

Lingyun Qiu[1], Rong Wu[2], Yinan Chen[3], Haixin Xiang[4], Wenming Zhan[1], Yinghao Zhang[1], Qiang Li[1], Huaxin Liu[1], Jieni Ding[1], Yucheng Li[1], Aihong Bi[1], Limin Luo[1], Yongshi Jia[1], Weijun Chen[1], Kainan Shao[1]*

1 Cancer Center, Department of Radiation Oncology, Zhejiang Provincial People's Hospital, Affiliated People's Hospital, Hangzhou Medical College, Hangzhou, Zhejiang, China, 2 Department of Breast Surgery, Hangzhou TCM Hospital Affiliated to Zhejiang Chinese Medical University, Hangzhou Hospital of Traditional Chinese Medicine, Hangzhou, Zhejiang, China., 3 School of Medical Technology, Yunnan Engineering Vocational College, Yunnan, China, 4 Radiotherapy and Chemotherapy Center, First Affiliated Hospital of Ningbo University, Ningbo, China

* shaokainan@hmc.edu.cn

**Editor:** Hesham M.H. Zakaly, Ural Federal University named after the first President of Russia B N Yeltsin Institute of Physics and Technology: Ural'skij federal'nyj universitet imeni pervogo Prezidenta Rossii B N El'cina Fiziko-tehnologiceskij institut, RUSSIAN FEDERATION

## Abstract

### Objective

Recurrent and metastatic tumors of the head and neck pose significant treatment challenges due to their proximity to critical structures and prior radiation exposure. This study aimed to evaluate the consistency between preoperative and post-operative dosimetric parameters in CT-guided [3]D-printed noncoplanar template (3DPNCT)-assisted radioactive iodine-125 seed implantation (RISI).

### Methods

Twenty-six patients with recurrent or metastatic head and neck cancer were retro-spectively analyzed. Gross tumor volume (GTV) coverage and dosimetric parameters such as D90 (dose covering 90% of the GTV), conformity index (CI), and homoge-neity index (HI) were compared before and after implantation. The Shapiro-Wilk test was used to assess data normality.

### Results

There were no significant differences between pre- and postoperative D90, V100, V150, or CI values (P > 0.05). Bland–Altman analysis showed high agreement for key metrics.

**Data availability statement:** All relevant data are within the manuscript and its Supporting Information files.

**Funding:** Zhejiang Provincial Department of Education General Scientific Research Project under Grant No. Y202457103.

**Competing interests:** The authors have declared that no competing interests exist.

## Conclusions

3DPNCT-assisted RISI demonstrated accurate dose delivery and high reproducibility. This approach may enhance local control while minimizing radiation to organs at risk in complex head and neck anatomies. These results suggest that this technique has promising clinical applicability in complex head and neck cases; however, further validation through larger prospective studies is warranted to confirm long-term efficacy and safety.

## Introduction

Malignant tumors represent a major global public health concern. According to 2022 global cancer statistics, China ranks first worldwide in both newly diagnosed cancer cases and cancer-related deaths. Moreover, recurrent and metastatic head and neck tumors are sources of major clinical concern because of their poor response to previous treatments, the head and neck are considered complex anatomical structures, and tumors in this region are in close proximity to critical organs [1,2]. Radiotherapy remains one of the most comprehensive treatments for cancer, with brachytherapy receiving particular attention for its precise dose distribution and minimal invasiveness. Radioactive seed implantation (such as $^{125}$I) enables continuous low-dose γ-ray emission, which creates a high-dose gradient within the target area without damaging the surrounding normal tissues, making it a particularly suitable salvage treatment option for recurrent tumors [3,4].

$^{125}$I seed implantation has been validated and routinely performed as part of prostate cancer treatment since the 1990s, becoming a standard treatment option for early-stage prostate cancer [5]. However, head and neck tumors present unique challenges due to anatomical heterogeneity, fibrosis induced by prior radiotherapy, and dose limitations associated with external beam radiation, often leading to inaccurate implantation. As the success of conventional image-guided puncture techniques is influenced by operator experience, inexperienced operators may create dose "cold spots" or "hot spots" due to deviations in puncture location, thereby affecting treatment efficacy [6]. Studies have revealed that 3D-printed noncoplanar templates (3DPNCTs) allow for individualized needle trajectory planning, which facilitates multiangle punctures and effectively maintains the discrepancy between postoperative dose verification parameters (such as D90 and V100) and preoperative values within 5%, thereby significantly improving treatment precision [6,7]. Nonetheless, the effectiveness of this technique can be limited by patient movement, and tumor displacement may cause the preplanned needle trajectory to deviate, particularly in cases of deep or mobile lesions [8,2].

In recent years, both dosimetric validation and the clinical application of 3DPNCTs have improved. Multiple studies revealed that 3DPNCT-assisted radioactive seed implantation for recurrent metastatic supraclavicular cancer, locally recurrent rectal cancer, and malignant liver tumors yields a greater than 95% postoperative dose coverage rate (V100), with a lower than 5% complication rate [9–11]. Regarding head

and neck tumors, a prospective cohort study by Jiang et al. revealed that 3DPNCT-assisted CT-guided [125]I seed implantation reduced the D90 dose deviation from 15% to 6% and significantly prolonged the median progression-free survival (PFS) to 14.2 months [12]. Furthermore, the introduction of dosimetric optimization algorithms has enhanced target conformity in complex anatomical regions (such as the skull base and carotid sheath), reducing radiation exposure to organs at risk (such as the optic nerve and brainstem) [13,14].

While emerging technologies such as robotic-assisted implantation and adaptive radiotherapy offer enhanced precision and real-time adaptation to anatomical variations [15,16], they often require advanced infrastructure, extended procedural time, and specialized operator training. In contrast, 3D-printed noncoplanar templates offer a relatively low-cost, customizable solution that integrates seamlessly with conventional CT-guided workflows. Their adaptability to complex anatomical geometries and capacity for individualized needle trajectory planning make them particularly suitable for salvage treatments in the head and neck region. These practical advantages support the selection of 3DPNCTs in this study over more technologically intensive alternatives.

Despite these advancements, several limitations remain. First, most available studies focus on verifying dosimetric feasibility and lack large-scale comparisons of preoperative and postoperative dose consistency as well as long-term efficacy evaluations [17–19]. Second, research on biological dose optimization for tumor heterogeneity (e.g., hypoxic regions or necrotic cores) remains in the exploratory phase, and the application of the linear–quadratic (L–Q) model-based equivalent dose formula (EQD2) in seed implantation has not yet been standardized [20–22]. Additionally, the cost and requirement for multidisciplinary collaboration limit the clinical adoption of 3DPNCTs, particularly in resource-limited healthcare settings [2,7].

To address these limitations, this study aims to systematically evaluate the consistency between preoperative and postoperative dosimetric parameters in 3DPNCT-assisted [125]I seed implantation for head and neck tumors. Furthermore, it assesses the clinical feasibility and precision of this approach within a routine, resource-constrained radiotherapy environment, thereby providing evidence to support its broader clinical adoption.

We hypothesized that there would be no significant differences between the preoperative and postoperative dosimetric parameters (null hypothesis, $H_0$), indicating accurate execution of the implantation plan. Conversely, the alternative hypothesis ($H_1$) posits that significant discrepancies exist, reflecting procedural variability or anatomical constraints.

The specific dosimetric outcomes evaluated include D90, D100, V100, V150, V200, CI (conformity index), HI (homogeneity index), and EI (external index), alongside the total number of seeds implanted. Bland–Altman analysis is further used to assess the consistency of each parameter. These metrics provide a comprehensive evaluation of spatial dose accuracy, plan reproducibility, and clinical feasibility.

In this study, 3DPNCTs were used and small backup needle columns were placed at 5 mm intervals in the anterior–posterior and lateral directions of traditional implantation needle columns. This approach is designed to increase the accuracy of seed implantation, improve the stability of implanted seeds, and reduce the need for template repositioning due to interpolation errors during surgery (as shown in Fig 1B). By reviewing the preoperative and postoperative dosimetric parameters (including D90, V100, conformity index [CI], and homogeneity index [HI]), as well as the radiation dose to organs at risk, the precision and clinical applicability of CT-guided [125]I seed implantation using 3DPNCTs for recurrent or metastatic head and neck tumors were determined. Therapeutic effectiveness was not evaluated in this study.

## Materials and methods

### Patient data

The clinical data of 26 patients with recurrent or metastatic head and neck tumors who were treated at Zhejiang Provincial People's Hospital between January 2018 and December 2022 were retrospectively analyzed. All patients underwent 3D-printed noncoplanar template-assisted CT-guided [125]I seed implantation. This retrospective study included all patients who met the eligibility criteria within the five-year recruitment window. No a priori power calculation was performed, as

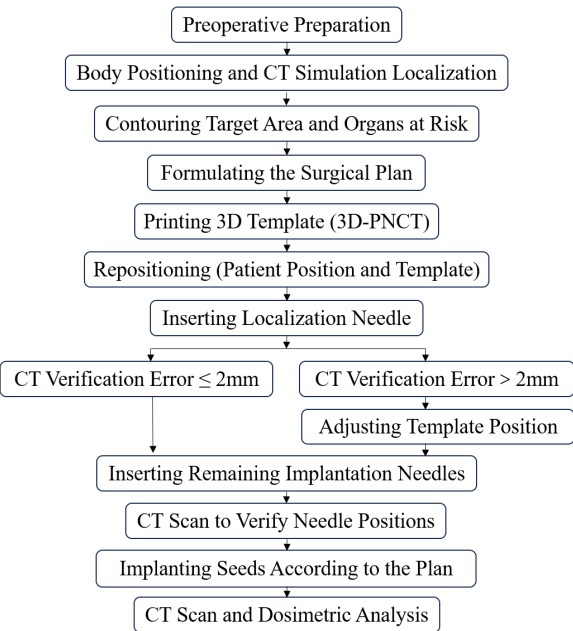

**Fig 1. Implant technical processes.**

the study was exploratory in nature and focused on assessing dosimetric reproducibility rather than hypothesis-driven clinical outcome comparison. Patients were eligible for inclusion if they met the following criteria: (1) age ≥ 18 years; (2) histologically confirmed recurrent or metastatic head and neck malignancy; (3) prior external beam radiotherapy (EBRT) was completed at least 6 months before seed implantation; and (4) availability of complete preoperative and postoperative CT imaging for dosimetric evaluation. Exclusion criteria included: (1) active bleeding or uncorrectable coagulopathy; (2) KPS < 60; (3) presence of distant metastases outside the head and neck region requiring systemic treatment priority; (4) incomplete dosimetric data or poor-quality imaging preventing reliable analysis.

The median age of the patients was 58 years (range: 31–80 years). Twenty-two patients were male, and 4 patients were female. The pathological types included nasopharyngeal carcinoma (13 patients), laryngeal carcinoma (4 patients), hypopharyngeal carcinoma (2 patients), maxillary sinus carcinoma (2 patients), thyroid carcinoma (2 patients), buccal mucosal carcinoma, floor of mouth carcinoma, and parotid gland carcinoma (1 patient each). The prescribed dose ranged from 90 to 120 Gy (median: 120 Gy), and the activity of [125]I seeds was 0.6 mCi (range: 0.5–0.7 mCi), as detailed in Table 1.

The retrospective study was approved by the Medical Ethics Committee of Zhejiang Provincial People's Hospital (No. 227 QT2024223) and was conducted in accordance with the ethical standards of the Declaration of Helsinki. Patient consent was waived by the Medical Ethics Committee of Zhejiang Provincial People's Hospital due to the anonymity of the data. All patient CT images were anonymized at the time of data collection. Data for this retrospective analysis were collected and analyzed between October and December 2024.

## Radioactive seed implantation procedure

To minimize procedural variability, the 3D-printed noncoplanar template-assisted CT-guided [125]I seed implantation technique is performed in accordance with a standardized precision-controlled workflow, as illustrated in Fig 1. All implantations were performed by one of two senior radiation oncologists and two medical physicists with over five years of

**Table 1. General characteristics of the 27 cases included in this review.**

|  | n |
| --- | --- |
| Gender |  |
| Male | 22 |
| Female | 4 |
| Age | 58 (31-80) |
| KPS Score | 80 (70-90) |
| Tumor stage |  |
| II | 3 |
| III | 13 |
| IV | 10 |
| Prescribed dose(Gy) | 120Gy (90Gy-120Gy) |
| Seed activity | 0.6 (0.5-0.7) |

experience in brachytherapy. Each operator had completed more than 50 3D-PNCT-guided implantation cases prior to the study period. Each step of the operation is detailed as follows:

1. Preoperative Preparation: A comprehensive patient assessment was performed, and all surgical instruments were prepared.

2. Patient Positioning and CT Simulation: A vacuum cushion was used for immobilization. The laser positioning system was calibrated so that the tumor center was the reference point, and both the patient and the vacuum cushion were marked. A CT scan with a 5 mm slice thickness (Discovery CT590, GE, Wisconsin, USA) was performed.

3. Target and Organ-at-Risk (OAR) Delineation: Image data saved in DICOM format were imported into the brachytherapy treatment planning system (BTPS, Vision7.3, Imaging Center of the Beijing University of Aeronautics and Astronautics, China) for delineation of the gross tumor volume (GTV) and OARs.

4. Preoperative Treatment Planning: The BTPS was used to design the noncoplanar implantation trajectory on the basis of tumor anatomy. The seed activity (0.6–0.8 mCi) and prescribed dose (PD) were set, ensuring a D90 ≥ 95% PD for the GTV and Organ-at-risk (OAR) dose constraints were determined based on published head and neck brachytherapy studies and the AAPM TG-43 formalism.[23].

5. 3D Template Fabrication: An SLA 3D printer and photosensitive resin were used to create a navigation template for each patient. Each 3D-PNCT has small auxiliary needle columns around the primary needle columns to enhance spatial registration.

The STL model for each 3D-printed template was generated based on the patient's CT scan (DICOM format). Segmentation of the skin surface, target tumor, and organs at risk was conducted using the BTPS. The planned needle entry points, angles, and depths were exported from the BTPS and incorporated into the template design using Mimics (Materialise, Belgium).

Spatial registration between CT and STL was maintained through two complementary strategies: (1) anatomical surface-matching, where the template undersurface was reverse-engineered to fit the patient's CT-derived body contour; and (2) fiducial-based alignment, using radiopaque markers placed on the patient during CT scanning, which were matched with physical grooves in the printed guide.

Virtual verification was performed using 3D Slicer to measure surface conformity and using CAD-based trajectory simulation to verify that needle channels aligned with planned paths. Physical accuracy was validated using a tissue-equivalent

phantom with embedded fiducials, and the needle positioning deviation was measured by postoperative CT fusion. Templates were approved for use only if deviations were ≤2 mm.

6.  Preoperative Repositioning and Calibration: Before needle insertion, the patient was immobilized, and the 3D-printed template was affixed to the skin surface. A reference needle was inserted through a predesignated trajectory. A validation CT scan (slice thickness 5 mm) was performed to assess the spatial alignment between the patient anatomy, template channels, and preoperative plan. If the deviation exceeded 2.0 mm, six-degree-of-freedom adjustments were applied for recalibration. Only when the alignment error was within acceptable limits (≤2.0 mm) did the procedure proceed.

7.  Verification of Needle Trajectories: During the procedure, after insertion of all or part of the implantation needles, a second CT scan was obtained to verify the actual needle trajectories. The spatial coordinates of each needle were compared with the planned path. If any deviation exceeded 2.0 mm, real-time adjustments were made. This step ensured that all needle trajectories matched the preplanned paths before radioactive seed delivery.

8.  Seed Implantation: [125]I seeds (0.6–0.8 mCi; 0.8 × 4.5 mm; nickel–titanium alloy) were implanted sequentially. All instruments used in the implantation procedure were disinfected by immersion in 2% glutaraldehyde solution at room temperature for no less than 10 hours, following standard high-level disinfection procedures for heat-sensitive medical devices.

9.  Immediate Postoperative Dosimetric Verification: The CT scan confirmed successful seed placement, and the dosimetric evaluation was performed before needle removal and hemostasis [24,25].

## Postoperative dosimetric verification and plan comparison

As shown in Fig 2, high-resolution CT images obtained immediately after surgery from a patient with recurrent parotid gland cancer (r-PGC) were imported into the BTPS. A hybrid registration algorithm was used for spatial mapping of the

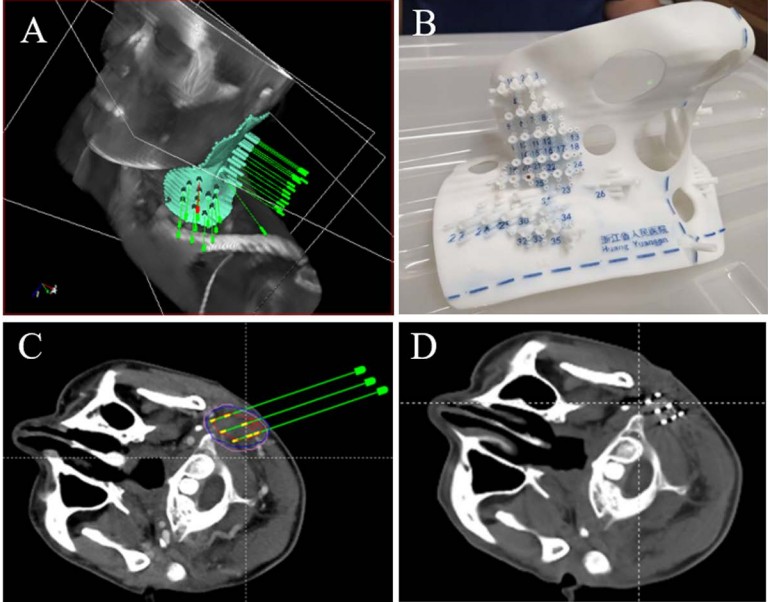

**Fig 2. The clinical operation of 3D-PNCT Assisted CT-Guided RISI for r-PGC. A** 3D-PNCT Design In TPS; **B** 3D-PNCT Design Physical Model; **C** Preoperative Planning: **D** Postoperative CT Scan.

preoperative plan onto the postoperative image, as it allowed rigid and deformable registration. Initially, deformable registration was applied to transfer the preoperative target and organ-at-risk (OAR) contours to the postoperative CT scan, with manual adjustments made as necessary. The width and level of the CT window were then adjusted to optimize visualization of the implanted seeds. The spatial positions of the implanted seeds were manually identified, and the postoperative target and OAR dosimetry values were subsequently calculated.

The target dose parameters included the minimum absorbed dose for 90% of the target volume (D90), the minimum absorbed dose for 100% of the target volume (D100), the percentage of the target volume covered by 100% of the prescription dose (V100), the percentage of the target volume covered by 150% of the prescription dose (V150), the percentage of the target volume covered by 200% of the prescription dose (V200), the conformal index (CI), the external index (EI), and the homogeneity index (HI). These parameters were evaluated in accordance with the recommendations described in the AAPM Task Group Report No. 43 (AAPM TG-43) [26].

A brief introduction to CI, EI, and HI is provided below:

**Conformity index (CI)**

The conformity index (CI) assesses the degree of alignment between the prescription dose region and the planned target volume (PTV) and therefore reflects the precision of dose coverage in the treatment plan. A CI value closer to 1 indicates greater conformity between the prescription dose and the target volume and therefore more precise dose coverage. The formula for CI is as follows:

$$CI = \frac{V_{RI}}{V_{PTV}}$$

where $V_{RI}$ represents the volume of the target region receiving the prescription dose or higher, and $V_{PTV}$ represents the volume of the planned target volume (PTV).

**External index (EI)**

The external index (EI) quantifies the extent to which the prescription dose extends beyond the target volume, reflecting the degree of high-dose exposure to normal tissues. A lower EI value indicates better dose confinement within the target and therefore less unnecessary radiation exposure to normal adjacent tissues. The formula for the EI is as follows:

$$EI = \frac{V_{RI} - V_{PTV}}{V_{PTV}}$$

where $V_{RI}$ represents the total volume receiving the prescription dose or higher, and $V_{PTV}$ represents the planned target volume.

**Homogeneity index (HI)**

The homogeneity index (HI) reflects the uniformity of the dose distribution within the target volume. An HI value closer to 0 suggests a more homogeneous dose distribution and therefore better treatment quality. The formula for HI is as follows:

$$HI = \frac{D_{2\%} - D_{98\%}}{D_{50\%}}$$

where $D_{2\%}$ represents the lowest dose received by 2% of the target volume, indicating the high-dose region. $D_{98\%}$ represents the lowest dose received by 98% of the target volume, indicating the low-dose region. $D_{50\%}$ represents the dose received by 50% of the target volume, serving as the median dose.

## Statistical analysis

The Shapiro–Wilk test was applied to assess the normality of the distribution of each dosimetric parameter. For normally distributed variables, the paired t-test was used; for non-normally distributed data, the Wilcoxon signed-rank test was used. A paired t test was conducted using SPSS 25.0 software to compare the preoperative and postoperative dosimetric parameters. A P value < 0.05 indicated statistical significance, whereas a P value > 0.05 indicated no statistically significant difference. Bland–Altman analysis was performed using MedCalc 20.0 software to evaluate the consistency of dosimetric parameters before and after surgery.

## Results

Twenty-six patients were included in this study, all of whom successfully underwent radiotherapy owing to precise target localization, which ensured accurate alignment between surface markings and the target tumor. A paired t test was conducted to compare the preoperative and postoperative dosimetric parameters and assess potential changes. The detailed results are presented in Table 2.

The mean number of implanted radioactive seeds was 47.9 preoperatively and 48.7 postoperatively, with no statistically significant difference (P > 0.05). This demonstrates a high level of consistency between planned and actual seed implantation, establishing a necessary foundation for subsequent dosimetric parameter comparisons.

### Comparison of target dosimetric parameters

Further analysis of preoperative and postoperative dosimetric parameters revealed no statistically significant differences (P > 0.05) in D90, V100, V150, V200, EI, or HI, indicating high consistency between the preoperative and postoperative dosimetric indices. However, significant differences (P < 0.05) were observed in D100 and CI, suggesting that these parameters may be influenced by intraoperative factors.

### Bland–Altman consistency analysis

The Bland–Altman method was employed to assess the consistency of the dosimetric parameters. The mean values of the preoperative and postoperative dosimetric parameters were plotted on the x-axis, and their differences were plotted on the y-axis. The 95% limits of agreement (confidence intervals, CIs) were described as the mean difference ± 1.96 standard deviations, and a Bland–Altman scatter plot was generated (Fig 3).

Specifically, D90: Mean difference = 18.59 Gy (95% CI: 9.7–27.5); 0% (0/27) exceeded the 95% limits of agreement, indicating good consistency. D100: Mean difference = 16.81 Gy (95% CI: 10.7–22.9); 0% (0/27) exceeded the 95% limits

**Table 2. Comparison of preoperative and postoperative validation parameters in 26 patients.**

| Parameter | Preoperative | | Postoperative | | P |
|---|---|---|---|---|---|
| | Interval | Mean ± Standard Deviation | Interval | Mean ± Standard Deviation | |
| Number of Seeds | 8-70 | 47.93 ± 18.53 | 5-79 | 48.71 ± 18.21 | 0.737 |
| $D_{90}$/Gy | 91.92-181.45 | 140.23 ± 22.91 | 85.60-175.74 | 134.68 ± 25.71 | 0.213 |
| $D_{100}$/Gy | 24.78-142.72 | 77.81 ± 22.71 | 28.73-98.06 | 61.55 ± 18.94 | 0.002 |
| $V_{100}$/% | 100.00-74.50 | 95.49 ± 5.65 | 80.90-99.50 | 92.56 ± 5.58 | 0.058 |
| $V_{150}$/% | 49.60-94.70 | 75.51 ± 10.80 | 54.50-93.30 | 70.15 ± 11.86 | 0.091 |
| $V_{200}$/% | 20.60-72.70 | 47.71 ± 13.14 | 27.90-80.80 | 48.58 ± 14.92 | 0.800 |
| CI | 0.38-0.73 | 0.54 ± 0.11 | 0.18-0.72 | 0.46 ± 0.15 | 0.008 |
| EI | 14.71-254.9 | 83.41 ± 52.30 | 16.34-211.15 | 119.61 ± 122.00 | 0.130 |
| HI | 5.28-38.45 | 21.21 ± 8.31 | 8.28-29.15 | 22.44 ± 8.86 | 0.610 |

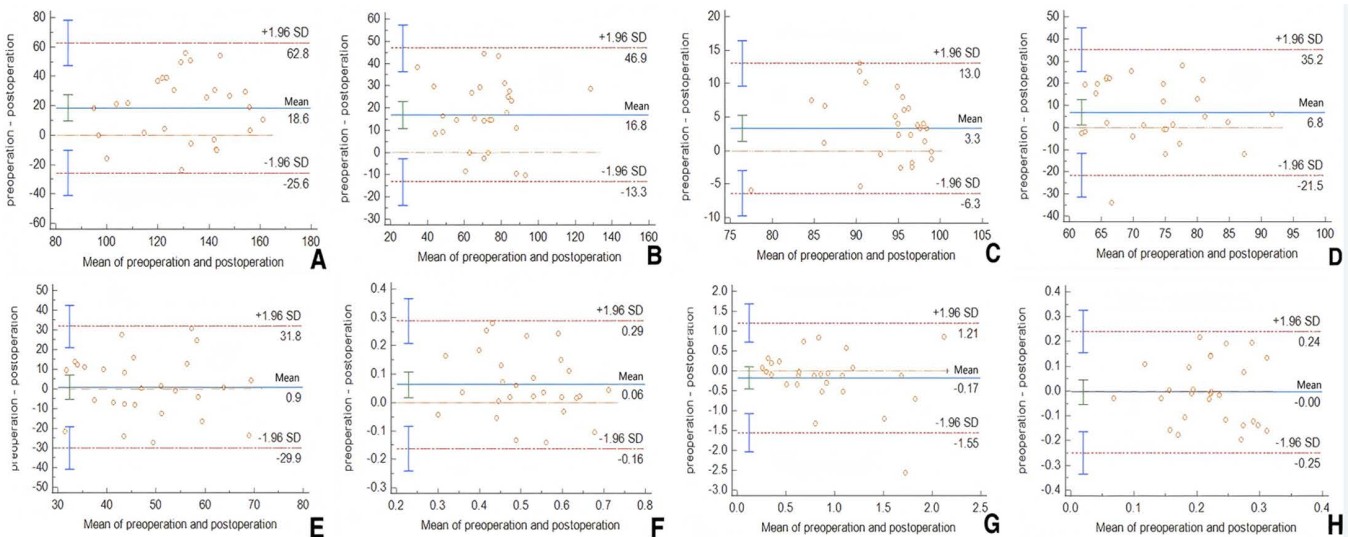

**Fig 3. Bland-Altman plot of differences in preoperative planning and postoperative dose parameters A $D_{90}$, B $D_{100}$, C $V_{100}$, D $V_{150}$, E $V_{200}$, F CI, G EI, H HI.**

of agreement, indicating good consistency. V100: Mean difference＝3.34 (95% CI: 1.4–5.3); 0% (0/27) exceeded the 95% limits of agreement, demonstrating good consistency. V150: Mean difference＝6.79 (95% CI: 1.1–12.6); 4% (1/27) fell below the 95% limits of agreement, suggesting poor consistency.

V200: Mean difference＝0.93 (95% CI: −5.3–7.1); 0% (0/27) exceeded the 95% limits of agreement, indicating good consistency. CI: Mean difference＝0.06 (95% CI: 0.02–0.11); 0% (0/27) exceeded the 95% limits of agreement, suggesting high consistency between the preoperative and postoperative measurements. EI: Mean difference＝−17.2% (95% CI: −45.1% to 10.6%); 4% (1/27) fell below the 95% limits of agreement, indicating poor consistency. HI: Mean difference＝−0.4% (95% CI: −5.4% to 4.5%); 0% (0/27) exceeded the 95% limits of agreement, demonstrating good consistency.

## Discussion

Previous studies have highlighted the technical limitations of freehand puncture in head and neck brachytherapy, including inconsistent needle angles and deviations that compromise dosimetric precision. Our study addresses these concerns by demonstrating that 3DPNCT-assisted implantation resulted in high consistency between pre- and postoperative D90, V100, and HI values. These findings confirm that template guidance effectively mitigates intraoperative variability, enhancing reproducibility and treatment accuracy in anatomically complex regions.

Therefore, ensuring that the planned seed position corresponds with the actual seed position remains a critical focus in research on seed implantation. Therefore, to overcome these challenges, a tool that can assist in precise needle placement while minimizing human error is needed.

Although the retrospective nature of this study introduces inherent limitations, we attempted to mitigate potential sources of bias by adhering to a highly standardized workflow and limiting procedural performance to a small group of experienced operators. This approach enhanced the internal consistency and reproducibility of the implantation procedures.

In recent years, 3D-printed noncoplanar templates (3D-PNCTs) has been recognized as a feasible solution for improving the accuracy of seed implantation. This template is useful for guiding needle placement, as it effectively overcomes

the limitations of traditional freehand puncture techniques, such as instability, low implantation accuracy, and a lack of quality control (QC) measures.

Our findings of high consistency between pre- and postoperative dosimetric parameters are consistent with previous studies. Qiu et al. reported that template-assisted implantation maintained D90 and V100 deviations within 5% of preoperative plans in head and neck cancers [6]. Similarly, Chen et al. demonstrated improved conformity indices and reduced hot spots using individualized 3D templates [7]. In contrast, studies using freehand or standard coplanar templates often reported larger deviations and higher operator dependency [27]. These differences highlight the role of template geometry, anatomical complexity, and procedural standardization in influencing dose accuracy.

In this study, the postoperative dosimetric parameters (D90, V100, EI, and HI) were not significantly different (P > 0.05) from the preoperative parameters, indicating that the 3D-printed noncoplanar template effectively ensures compliance with the preoperative dosimetric plan, thereby ensuring precise tumor coverage. Although the postoperative EI and HI values were slightly better than the preoperative values, the changes were not statistically significant, suggesting that this technique is reliable and consistent in controlling the dose distribution outside the target volume and maintaining dose homogeneity. These findings are consistent with those of previous studies [25,26], further validating the feasibility and efficacy of using 3D-printed noncoplanar templates in radioactive seed implantation for head and neck tumors. However, this study revealed that the pre- and postoperative D100 and CI values were significantly different (P < 0.05), suggesting that the postoperative dose distribution may still deviate to some extent from the preoperative distribution. Possible explanations include the following:

Minor errors in patient positioning: Although vacuum cushions and other fixation devices were used intraoperatively to increase positioning reproducibility, slight positioning errors may still have occurred, leading to subtle variations in needle direction and depth. Needle deviation due to tissue resistance: During needle insertion, the implantation needle must pass through various tissues and bony structures, where nonuniform resistance may cause slight needle deformation, leading to positional deviations. Image registration errors between preoperative and postoperative scans: The time interval between preoperative and postoperative CT scans may result in slight tissue shifts, affecting the precise alignment of the target volume and thereby introducing minor variations in dose distribution.

Additionally, Bland–Altman consistency analysis revealed good agreement for D90, D100, V100, V200, CI, and HI and poor consistency of V150 and EI. The discrepancies in V150 and EI may be attributed to the following:

Optimization of seed implantation during surgery: In some cases, dose optimization is performed intraoperatively, leading to a different number of implanted seeds and therefore differences between the preoperative and postoperative dosimetric parameters.

Anatomical changes in the patient: Owing to the time interval between preoperative and postoperative CT scans, there may be subtle anatomical changes that affect target volume consistency and lead to minor differences in dose distribution.

In clinical practice, seed migration or loss may occur after implantation. In this study, immediate postoperative CT revealed no seed displacement. However, tissue interactions and patient movement may still contribute to postoperative seed migration. To reduce this risk, seeds can be implanted in a chain link configuration. Additionally, regular CT follow-ups are recommended to monitor seed stability. Although no clinically significant seed migration was detected during routine follow-up in this cohort, systematic analysis was not performed. Future prospective tracking and correlation with published migration rates are warranted.

Although the prescribed doses ranged from 90 to 120 Gy, no subgroup comparison was performed to assess whether higher doses improved GTV coverage (e.g., V100). This was due to limited statistical power in the current sample. In future studies with larger cohorts, we plan to stratify patients by prescription dose and use ANOVA or Kruskal–Wallis testing to evaluate dose-response effects.

An important limitation of this study is the inclusion of patients with diverse tumor types and anatomical locations. Variations in tumor depth, local tissue compliance, and adjacency to organs at risk may influence needle placement accuracy

and, consequently, dosimetric consistency. While this heterogeneity mirrors routine clinical practice, it may obscure location-specific performance differences of the 3D-printed template system. Future studies with larger cohorts should consider stratified or subgroup analyses to evaluate anatomical factors affecting implantation precision. While this study confirmed the dosimetric precision of 3DPNCT-assisted implantation, it did not evaluate clinical endpoints such as tumor response or patient survival. Future prospective studies with structured follow-up are necessary to correlate dosimetric parameters with long-term oncologic outcomes and treatment-related toxicity profiles.

The small sample size may have affected the statistical stability of the findings. A larger sample of patients are needed in future studies to validate these results. The 3D template is reconstructed from preoperative CT data, but because of delays in production, subtle anatomical changes may be observed, leading to minor mismatches between the template and anatomical structures intraoperatively. Looking forward, the integration of rapid prototyping technologies may significantly reduce the fabrication time of 3D-printed templates from several hours to under 30 minutes, enabling near real-time production and potential intraoperative use [28].

Stratified analyses were not performed due to limited sample size, potentially masking subgroup-specific differences. Long-term safety assessments were not conducted, limiting conclusions about persistent seed migration risks. Measurement bias from potential image fusion errors between pre- and postoperative scans could affect dosimetric accuracy. The retrospective design and convenience sampling introduce potential selection bias. Finally, caution is necessary in interpreting the clinical applicability of these promising results, given the current absence of conclusive clinical outcomes. Despite promising dosimetric results, definitive clinical applicability requires conclusive clinical data from future prospective studies.

## Conclusion

This study demonstrated that 3D-printed noncoplanar template-assisted CT-guided $^{125}$I seed implantation achieves high dosimetric consistency between preoperative plans and postoperative outcomes in patients with recurrent or metastatic head and neck tumors. The technique enhances implantation accuracy, reduces intraoperative variability, and supports safe dose delivery in anatomically complex regions. However, the relatively small sample size and the potential influence of anatomical shifts on template alignment highlight the need for further validation in larger, prospective studies. Future research should explore the integration of intraoperative imaging, real-time adaptive planning, and AI-guided needle placement to further minimize deviations and optimize therapeutic outcomes.

## Supporting information

**S1 Data. Rawdata_preoperative.** Raw data part 1 as csv.
(CSV)

**S2 Data. Rawdata_postoperative.** Raw data part 2 as csv.
(CSV)

## Acknowledgments

Thanks to American Journal Expert for help with language editing.

## Author contributions

**Conceptualization:** Rong Wu, Wenming Zhan.

**Data curation:** Haixin Xiang.

**Formal analysis:** Yinghao Zhang.

**Funding acquisition:** Qiang Li.

**Investigation:** Huaxin Liu.

**Methodology:** Jieni Ding.

**Project administration:** Weijun Chen.

**Resources:** Yucheng Li.

**Software:** Yinan Chen.

**Supervision:** Aihong Bi, Yongshi Jia.

**Visualization:** Limin Luo.

**Writing – original draft:** Lingyun Qiu.

**Writing – review & editing:** Kainan Shao.

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
