## [Decision Letter · Decision Letter 0]

7 May 2025

Dear Dr. Shao,

Thank you for submitting your manuscript to PLOS ONE. After careful consideration, we feel that it has merit but does not fully meet PLOS ONE’s publication criteria as it currently stands. Therefore, we invite you to submit a revised version of the manuscript that addresses the points raised during the review process.

We look forward to receiving your revised manuscript.

Kind regards,

Hesham M.H. Zakaly, Ph.D.

Academic Editor

PLOS ONE

Journal Requirements:

This research was supported by Zhejiang Provincial Department of Education General Scientific Research Project under Grant No. Y202457103. Thanks to American Journal Expert for help with language editing.

Zhejiang Provincial Department of Education General Scientific Research Project under Grant No. Y202457103

4. Please remove all personal information, ensure that the data shared are in accordance with participant consent, and re-upload a fully anonymized data set.

Reviewers' comments:

Reviewer's Responses to Questions

**Comments to the Author**

1. Is the manuscript technically sound, and do the data support the conclusions?

Reviewer #1: Yes

Reviewer #2: No

Reviewer #3: Yes

Reviewer #4: Yes

2. Has the statistical analysis been performed appropriately and rigorously?

Reviewer #1: N/A

Reviewer #2: No

Reviewer #3: Yes

Reviewer #4: Yes

3. Have the authors made all data underlying the findings in their manuscript fully available?

Reviewer #1: Yes

Reviewer #2: Yes

Reviewer #3: Yes

Reviewer #4: Yes

4. Is the manuscript presented in an intelligible fashion and written in standard English?

Reviewer #1: Yes

Reviewer #2: Yes

Reviewer #3: Yes

Reviewer #4: Yes

Reviewer #1: Abstract

1- The abstract would benefit from a brief background sentence to set the context and explain why evaluating 3D-printed template-assisted implantation is clinically relevant in head and neck cancers.

2- The conclusion could be slightly expanded to comment on clinical significance or suggest the need for further validation in larger or prospective studies.

Introduction

Overall, the introduction does a good job of providing strong background context, clearly outlining the burden of head and neck cancers, the challenges in treating them, and the advantages of using ¹²⁵I seed implantation. It effectively explains the rationale for using 3D-printed noncoplanar templates (3DPNCTs) and references prior work and technological improvements that have contributed to this field. However, there are several areas where the introduction could be strengthened to improve clarity and scientific rigor. Below are some points to consider:

1- I think the introduction does a good job showing the benefits of 3D-printed noncoplanar templates and backing that up with previous studies. But it would be helpful if the authors briefly compared this technique with other newer approaches, like robotic-assisted implantation or adaptive radiotherapy. That would make it clearer why 3DPNCTs were chosen and what makes them stand out in this context.

2-

“Despite these advancements, several limitations remain. First, most available studies focus on verifying dosimetric feasibility and lack large-scale comparisons of preoperative and postoperative dose consistency as well as long-term efficacy evaluations[16–18]. Second, research on biological dose optimization for tumor heterogeneity (e.g., hypoxic regions or necrotic cores) remains in the exploratory phase, and the application of the linear‒quadratic (L‒Q) model-based equivalent dose formula (EQD2) in seed implantation has not yet been standardized[19–21]. Additionally, the cost and requirement for multidisciplinary collaboration limit the clinical adoption of 3DPNCTs, particularly in resource-limited healthcare settings”

That paragraph lists several important limitations in the current literature:

• Lack of large-scale comparisons between pre-op and post-op dosimetry

• Limited research on biological dose optimization for heterogeneous tumors

• No standardized use of EQD2 formulas

• Practical limitations like cost and need for collaboration

These are valid and well-described gaps in the field.

But the issue is: the authors don’t clearly say how their own study will address those specific gaps.

For example, they could have followed that paragraph with something like: (To address these issues, this study aims to evaluate dosimetric consistency between pre- and postoperative plans and assess clinical feasibility in a resource-limited setting)

3- The study lacks a clear mention of the null and alternative hypotheses, which are essential for understanding the specific questions the authors aimed to address. Additionally, the introduction provides valuable background, but it would be helpful to explicitly state the study’s aim and the expected outcomes. Could the authors clarify the hypotheses and the specific outcomes they intend to measure to strengthen the study’s scientific framework?.

Patients and methods

“The clinical data of 26 patients with recurrent or metastatic head and neck tumors who were treated at Zhejiang Provincial People's Hospital between January 2018 and December 2022 were retrospectively analyzed”.

1- How was the sample size of 26 patients determined? Did the authors perform a power analysis before the study or sample size justification? If so, which power and effect size did they use to calculate the sample size?

2- What were the inclusion and exclusion criteria for selecting the patients in this study? Clear criteria help in understanding the population studied and the relevance of the findings.

3- What method was used for retrospective patient selection? Were the patients selected based on specific clinical characteristics, diagnostic criteria, or treatment history?

4- In the methodology section, you mention that the study includes 26 patients, but the gender and pathological type breakdown adds up to 27 patients. The details are as follows:

• Gender: 23 male patients, 4 female patients (Total: 23 + 4 = 27 patients)

Pathological types of head and neck cancers:

Nasopharyngeal carcinoma: 14 patients

Laryngeal carcinoma: 4 patients

Hypopharyngeal carcinoma: 2 patients

Maxillary sinus carcinoma: 2 patients

Thyroid carcinoma: 2 patients

Buccal mucosal carcinoma: 1 patient

Floor of mouth carcinoma: 1 patient

Parotid gland carcinoma: 1 patient

(Total: 14 + 4 + 2 + 2 + 2 + 1 + 1 + 1 = 27 patients)

Please clarify the discrepancy between the total number of patients (26) mentioned earlier in the text and the breakdown provided here, which adds up to 27. Adjusting for consistency throughout the methodology section is recommended.

5- I noticed that the study includes patients with a variety of tumor types, which may differ in size, location, blood supply, and tissue depth. I wonder if there was any consideration to standardizing the patient group by tumor type or location to better assess the effectiveness of the 3D-printed noncoplanar template-assisted implantation technique in specific areas. It would be interesting to see if the dosimetric outcomes varied across different tumor characteristics, or if this variation could influence the overall findings.

6- The study design is retrospective, which is common in dosimetric research. However, this approach comes with some limitations. One main issue is that, since the study looks at past data, the researchers are unable to control for all factors that could affect the results (such as patient characteristics or variations in how the procedures were performed).

It would be helpful if the authors could clarify how they minimized potential biases in this study. For example, did they ensure that all patients followed standardized procedures? Were the same operators consistently performing the procedures? Addressing these points would strengthen the reliability of the results and help mitigate any potential biases.

7- While the authors state that each step of the implantation workflow is detailed, the process of designing the 3D-printed noncoplanar template remains unclear in some technical aspects.

- How the STL file was created (from what input - e.g., segmentation, external surface scanning, etc.)

- How the spatial registration between the CT scan (DICOM) and the 3D template (STL) was performed.

- How the alignment between virtual planning and printed guide was verified before use

- What software was used for merging or verifying this integration

8- A reader or another institution cannot easily reproduce the procedure without:

o Knowing the BTPS software used

o Understanding how trajectory planning was translated to physical needle guide paths

9- Steps 6 and 7 seem to describe very similar things — both talk about using a CT scan to check if the needles are in the right place and making sure the error is less than 2 mm. It’s a bit hard to tell if these are two different steps or just two ways of explaining the same thing. It might be clearer if the authors either combine them or explain more clearly how they’re different

“6- Preoperative Repositioning and Calibration: The patient was repositioned and then secured. A reference needle was inserted, and a validation CT scan (5 mm slice thickness) was performed. If the target registration error was ≤ 2.0 mm, implantation proceeds; if the target registration error was > 2.0 mm, a six-degree-of-freedom model was used for recalibration.

7. Verification of Needle Trajectories: The CT scan ensured proper needle placement, with corrections made until errors were ≤ 2 mm.”

10- “The instruments were sterilized with wet heat.”

The statement that instruments were sterilized using 'wet heat' is somewhat vague. It would be helpful to specify the exact sterilization method - for example, whether steam autoclaving was used, and if so, under what parameters (e.g., temperature and duration). Providing these details would improve clarity and reproducibility of the protocol.

11- Preoperative Treatment Planning: The BTPS was used to design the noncoplanar implantation trajectory on the basis of tumor anatomy. The seed activity (0.6–0.8 mCi) and prescribed dose (PD) were set, ensuring a D90 ≥ 95% PD for the GTV and OAR doses within acceptable limits

The dose constraints for OARs are said to be “within acceptable limits” but it would be more rigorous to clearly state whether this follows an established guideline.

Discussion

8- The discussion includes a detailed explanation of the limitations of freehand puncture techniques in head and neck brachytherapy, which is important background. However, this information was already covered in the introduction. To avoid redundancy, the authors may consider summarizing this section more briefly or rephrasing it to focus specifically on how their findings support or address these previously mentioned challenges.

9- The brief note on seed migration is appreciated, but it may be helpful to elaborate on follow-up duration and whether the study design allowed for delayed migration to be captured. A comparison with seed migration rates in the literature would also provide useful context.

10- The authors mention that integration of rapid prototyping could improve future workflows, which is promising. Expanding slightly on how automation or software advancements (e.g., intraoperative re-planning or AI-based registration) might reduce remaining sources of error would enhance the forward-looking part of the discussion.

11- The discussion would benefit from referencing previous studies to support or contrast the current findings. Including citations of studies that have shown similar results would help validate the conclusions, while mentioning studies with different outcomes could offer a deeper understanding of the variables that influence implantation accuracy and dose distribution.

Additionally, the manuscript would be strengthened by a more forward-looking perspective. The current mention of 'rapid prototyping' is a good start, but future research directions could include exploring real-time image-guided implantation, intraoperative adaptive planning, automated needle path planning, or AI-assisted registration systems. These advancements could further minimize deviations and improve accuracy in complex anatomical regions like the head and neck.

Conclusions

1- The conclusion restates key findings effectively, but it could be more concise by avoiding repetition of earlier points already addressed in the discussion. And also would benefit from briefly acknowledging limitations, such as the small sample size or anatomical shifts that could influence template alignment. Including these would give a more balanced and realistic summary.

2- The authors may also consider adding a sentence about future research directions, such as integrating intraoperative imaging, real-time adaptive planning, or AI-guided needle placement to further reduce intraoperative deviations and improve outcomes

Reviewer #2: First, I would like to thank you for the invitation to review the study “Dosimetry study of 3D-printed noncoplanar template-assisted CT-guided 125I seed implantation for the treatment of recurrent and metastatic tumors in the head and neck.” Through this evaluation, I will seek to contribute with well-founded opinions for the advancement of science and the improvement of humanity’s quality of life.

Considering initially "only the abstract of the article," this is important because it will either encourage or discourage a reader’s intention to proceed to a full reading of the manuscript. Thus, I make the following observations regarding this section:

The text used several acronyms without explaining them at their first mention;

The text focused only on dosimetric precision but did not mention real clinical effectiveness objectives, such as patient survival for each dose level, limiting the clinical interpretation of the findings and potentially generating disinterest among future readers;

The statistical analysis has positive points but also presents flaws, such as the lack of reporting on normality testing and a brief justification for the sample size (if space for addition or editing is possible).

Regarding the main body of the manuscript, I make the following observations: I sought to evaluate the present retrospective observational study based on the application of the NOS (Newcastle–Ottawa Scale), aiming to be rigorous using a widely recognized metric in the scientific community.

Regarding the selection process: Concerning the representativeness of the sample, the study involved 26 patients from a single hospital.

The sample was one of convenience and limited in size, with no sample size calculation or adequate statistical justification.

Regarding the selection of the non-exposed cohort, the study was single-armed, with no control group.

Regarding the ascertainment of exposure, it was adequate. The procedure was well described (CT and 3D template). Regarding the outcome not being present at the start of the study, it was appropriate for dosimetry purposes; however, the clinical outcome of oncological effectiveness, such as survival or mortality rates, unfortunately, was not the focus.

In the context of the representativeness of the exposed cohort, according to the NOS, 2 out of 4 stars were awarded, that is, 50% of the possible points in this domain.

Regarding the control of confounding factors, no formal control was observed, as there was no comparative group or multivariate analysis.

Thus, in the context of comparability, according to the principles of the NOS, no stars were awarded, resulting in no score in this quality domain.

The outcome assessment was based on objective measures (dosimetric parameters such as D90, V100...), and not on subjective evaluations.

Regarding the follow-up time, it was not subject to evaluation, as the focus was only on immediate dosimetry, unfortunately not addressing long-term clinical outcomes.

Regarding the completeness of follow-up, all patients were evaluated post-procedure.

In the context of outcome assessment, according to the NOS, 2 out of 3 stars were awarded, corresponding to 66.7% of the possible points in this domain.

The final NOS score was four out of nine possible stars, yielding a performance of 44.44%, indicating a moderate level of bias risk but with important limitations.

Identification of biases in the study:

Selection bias: Small sample size and single-center recruitment.

Bias due to absence of control group: No comparison between techniques, such as template-guided versus freehand implantation.

Measurement bias: Although objective tests (CT, dosimetry) were used, the exclusive use of dose parameters without clinical outcomes limits inferences regarding real therapeutic effectiveness.

Confounding bias: Heterogeneity of tumors, involving different types, stages, and anatomical locations, without statistical adjustment.

The purpose of treatment effectiveness: The study evaluated dosimetric precision (alignment between planning and execution), without evaluating clinical effectiveness (control over the progression of the malignant neoplasm, survival, and/or mortality).

Thus, the title “...treatment of recurrent and metastatic tumors…” can be interpreted as referring to technical execution, but it is not possible to conclude on the real oncological benefit for the patient. No data were presented on: Tumor response rates; Disease progression; Overall survival or progression-free survival; Long-term complications.

Heterogeneity of the sample:

Regarding tumor types: Nasopharynx (14), larynx (4), hypopharynx (2), maxillary sinus (2), thyroid (2), buccal mucosa, floor of the mouth, and parotid gland (1 each). A relevant problem with this variety is the significant biological diversity among tumors and different responses to radiotherapy.

Regarding disease stages: Stage II (3 patients), III (13 patients), and IV (10 patients). Here, the problem lies in the varied expectations for local control and survival among the different stages.

Regarding anatomical tumor locations: Tumors located in areas with different anatomical risks, for example, near nerves and/or blood vessels, which may impact the precision, safety, and effectiveness of interventions.

This heterogeneity compromises the internal validity of the study, as without stratification, it is not possible to determine whether the technique would be equally effective and safe for all types of neoplasms evaluated.

Statistical Analysis:

Regarding the sample size calculation: It was not performed, and no justification was provided.

Normality testing: Not mentioned; however, the authors used the paired t-test, which presupposes prior testing for data normality.

Pre- and postoperative comparison: Performed using the paired t-test or should it have been the Wilcoxon test? We could only know after verifying the normality of the data distribution.

Regarding the comparison among multiple dosages: The study only compared before and after, without comparing different dose subgroups (90–120 Gy). I missed the opportunity for the authors to explore whether higher doses would lead to better coverage (V100) or not. However, it would have been necessary to use ANOVA or the Kruskal-Wallis test, depending on the data distribution (normal or not), but unfortunately, this curiosity was not pursued.

Finally, the authors present a good study idea regarding dosimetry using 3D templates, but their methodological limitations, especially the lack of clinical effectiveness assessment and control of heterogeneity, prevent the technique from being broadly generalized as effective for the treatment of head and neck malignancies.

Reviewer #3: Well thought out.

Appropriate statistics.

Good explanation of methodology.

Minimal discussion on limitations, however may be appropriate for this type of proof of concept and will need further assessment.

Reviewer #4: More number of patients and combined studies should be considered in future in such a study.Such studies can help in further making brachytherapy more precise as we tend to eliminate human error of free puncture.

**Do you want your identity to be public for this peer review?** For information about this choice, including consent withdrawal, please see our Privacy Policy

Reviewer #1: **Yes: ** Safwat Elwaseef

Reviewer #2: No

Reviewer #3: No

Reviewer #4: No

Responses_to_reviewers_PLOSONE.docx

---

## [Author Response · Author response to Decision Letter 1]

23 May 2025

Included in the attachment file 'Responses_to_reviewers_PLOSONE.pdf'.

---

## [Decision Letter · Decision Letter 1]

8 Jun 2025

Dear Dr. Shao,

Thank you for submitting your manuscript to PLOS ONE. After careful consideration, we feel that it has merit but does not fully meet PLOS ONE’s publication criteria as it currently stands. Therefore, we invite you to submit a revised version of the manuscript that addresses the points raised during the review process.

We look forward to receiving your revised manuscript.

Kind regards,

Hesham M.H. Zakaly, Ph.D.

Academic Editor

PLOS ONE

Journal Requirements:

Reviewers' comments:

Reviewer's Responses to Questions

**Comments to the Author**

Reviewer #1: All comments have been addressed

Reviewer #2: (No Response)

2. Is the manuscript technically sound, and do the data support the conclusions?

Reviewer #1: Yes

Reviewer #2: Partly

3. Has the statistical analysis been performed appropriately and rigorously?

Reviewer #1: Yes

Reviewer #2: Yes

4. Have the authors made all data underlying the findings in their manuscript fully available?

Reviewer #1: Yes

Reviewer #2: Yes

5. Is the manuscript presented in an intelligible fashion and written in standard English?

Reviewer #1: Yes

Reviewer #2: Yes

Reviewer #1: I have carefully reviewed the manuscript and have no concerns regarding dual publication, research ethics, or publication ethics. I find the work acceptable in its current form and have no further comments for the author.

Reviewer #2: "Dosimetry study of 3D-printed noncoplanar template-assisted CT-guided 125I seed implantation for the treatment of recurrent and metastatic tumors in the head and neck"

Considering the comments made in the first review of the manuscript, I observe that there were improvements in some aspects and lack of improvement in others.

Improvements implemented:

The research question was clearly presented at the end of the introduction, appropriately separating it from the methods section.

The eligibility criteria are now explicitly described, including age, tumor type, history of radiotherapy, and clinical conditions.

All dosimetric variables were defined with clear units (Gy, %) in the methods and results.

The study was correctly characterized as a retrospective study with a dosimetric method, without confusing it with a "meta-analysis" as a study type.

Although there was no explicit mention of language/year restrictions, the study is retrospective, based on hospital patient data (2018–2022), which implies a clear temporal restriction.

Problems still observed:

Small and heterogeneous sample: A sample of 26 patients with multiple tumor types and anatomical sites, making generalizations difficult. (It is recommended that this be noted in the discussion, before the final conclusion, as a limitation of the study).

Lack of stratified analysis.

Absence of clinical outcomes – Local control, survival, or toxicity were not assessed.

Lack of comparison with other techniques – No comparison with manual technique, coplanar guides, or robotic approaches.

Limited safety assessment – Only immediate evaluation of seed migration, without follow-up.

Selection bias: Retrospective, convenience sampling (It is recommended that this be noted in the discussion, before the final conclusion, as a limitation of the study).

Measurement bias – Image fusion subject to errors (It is recommended that this be noted in the discussion, before the final conclusion, as a limitation of the study).

Operator bias – Performed by a highly specialized team, limiting generalization (should be highlighted to the clinical community).

Possibly Overestimated Results.

Partial dosimetric consistency: D100 and CI showed significant differences.

Optimistic clinical applicability: Are there conclusive clinical data?

Implant stability not proven – No follow-up for late migration (It is recommended that this be noted in the discussion, before the final conclusion, as a limitation of the study).

The study presents promising results but requires methodological improvements for clinical validation and broader adoption, clearly reporting the limitations that must be considered when thinking about a more generalized use

**Do you want your identity to be public for this peer review?** For information about this choice, including consent withdrawal, please see our Privacy Policy

Reviewer #1: No

Reviewer #2: No

---

## [Author Response · Author response to Decision Letter 2]

9 Jun 2025

Included in the file 'Responses_to_reviewers_PLOSONE.docx".

---

## [Decision Letter · Decision Letter 2]

30 Jul 2025

Dosimetry study of 3D-printed noncoplanar template-assisted CT-guided 125I seed implantation for the treatment of recurrent and metastatic tumors in the head and neck

PONE-D-25-15916R2

Dear Dr. Shao,

We’re pleased to inform you that your manuscript has been judged scientifically suitable for publication and will be formally accepted for publication once it meets all outstanding technical requirements.

Kind regards,

Hesham M.H. Zakaly, Ph.D.

Academic Editor

PLOS ONE

Additional Editor Comments (optional):

Reviewers' comments:

Reviewer's Responses to Questions

**Comments to the Author**

Reviewer #2: All comments have been addressed

2. Is the manuscript technically sound, and do the data support the conclusions?

Reviewer #2: Yes

3. Has the statistical analysis been performed appropriately and rigorously?

Reviewer #2: Yes

4. Have the authors made all data underlying the findings in their manuscript fully available?

Reviewer #2: Yes

5. Is the manuscript presented in an intelligible fashion and written in standard English?

Reviewer #2: Yes

Reviewer #2: As this is my third review, I will be more objective in informing you that, considering the addition of information about the article's limitations in the discussion, I understand that the authors made it more suitable.

**Do you want your identity to be public for this peer review?** For information about this choice, including consent withdrawal, please see our Privacy Policy

Reviewer #2: No

---

## [Editor Report · Acceptance letter]

PONE-D-25-15916R2

PLOS ONE

Dear Dr. Shao,

I'm pleased to inform you that your manuscript has been deemed suitable for publication in PLOS ONE. Congratulations! Your manuscript is now being handed over to our production team.

Kind regards,

on behalf of

Dr. Hesham M.H. Zakaly

Academic Editor

PLOS ONE